# Candidate correlates of protection in the HVTN505 HIV-1 vaccine efficacy trial identified by positive-unlabeled learning

Shiwei Xu[1], Aaron Hudson[2], Holly E. Janes[2], Georgia D. Tomaras[3], Margaret E. Ackerman [1,4,5]*

1 Quantitative Biomedical Sciences Program, Dartmouth College, Hanover, New Hampshire, United States of America, 2 Biostatistics, Bioinformatics and Epidemiology Program, Vaccine and Infectious Disease Division, Fred Hutch Cancer Center, Seattle, Washington, United States of America, 3 Department of Surgery, Center for Human Systems Immunology, Duke University, Durham, South Carolina, United States of America, 4 Department of Microbiology and Immunology, Geisel School of Medicine at Dartmouth, Dartmouth College, Hanover, New Hampshire, United States of America, 5 Thayer School of Engineering, Dartmouth College, Hanover, New Hampshire, United States of America

* margaret.e.ackerman@dartmouth.edu

## Abstract

With a goal of unveiling mechanisms by which vaccines can provide protection against HIV-1 acquisition, several studies have explored correlates of risk of HIV-1 acquisition in HVTN 505, which was a phase IIb trial conducted to assess the safety and efficacy of a DNA plasmid and recombinant adenovirus serotype 5-vectored HIV vaccine regimen among individuals in the United States who were vulnerable to acquiring HIV. While this trial failed to meet its predetermined efficacy criteria, both immunological and virological correlates of reduced risk of acquisition have been reported, suggesting that at least some vaccine recipients were protected from some viruses. In this work, we describe application of a novel Positive-Unlabeled machine learning-based approach to infer protection status among vaccine recipients that did not acquire HIV, resulting in improved power to detect potential correlates of immunity. Having established the analytical robustness of protection status predictions using cross-validation and permutation testing strategies, we report increased confidence in previously identified correlates of risk, such as vaccine-elicited anti–HIV-1 Env glycoprotein IgG3 antibodies and antibody-dependent phagocytosis, and the new observation of an inverse correlation between inferred vaccine-mediated protection and virus-specific IgA responses. Though its biological validity is not established, this inference approach offers a new means to use case-control datasets to identify candidate markers of effective immune responses in the context of low vaccine efficacy.

**Data availability statement:** The datasets used in this analysis are available at https://atlas.scharp.org/project/HVTN%20Public%20Data/HVTN%20505/begin.view. The code used in this work is available at: https://github.com/AckermanLab/Xu_et_al_HVTN505.

**Funding:** This study was supported in part by NIAID R56AI165448 to M.E.A. and P01AI162242 to M.E.A. and G.D.T and by the HVTN505 study team NIH 5UM1AI068618 to G.D.T. The funders had no role in study design, data collection and analysis, decision to publish, or preparation of the manuscript.

**Competing interests:** I have read the journal's policy and the authors of this manuscript have the following competing interests: M.E.A. reports consulting for Seromyx Systems and research funding unrelated to HIV vaccines from Moderna.

## Author summary

Field trials of vaccine efficacy are important to identifying immunization regimens that prevent infection or reduce disease severity. Such studies can also be used to identify correlates of protection, which are used in vaccine licensure and help us understand potential mechanisms of protection. However, unlike studies in which participants are directly challenged with a pathogen, field trials rely on real-world exposure to pathogens that may or may not occur for all participants during the study, which reduces our ability to discover such correlates. This work describes a method whereby immune response profiles are used to infer whether study participants that were not infected were more likely to have been protected or simply not exposed to pathogen, which can improve our ability to discover potential correlates and learn about attributes of immune responses that may help protect from infectious disease.

## Introduction

One challenge to development of an efficacious HIV-1 vaccine is a lack of clear and reproducible immune correlates of protection, measures or traits that definitively relate to protection from HIV acquisition, which serve to provide understanding of the required immune response to prevent HIV acquisition, and can be reliable biomarkers to predict vaccine efficacy in humans [1]. Among vaccine efficacy trials conducted to date, RV144 (NCT00223080) is the only clinical trial to meet predetermined end-point efficacy criteria, demonstrating a moderate efficacy of 31.2% (95% confidence interval, 1–51%) at three years [2]. In contrast, in 2013, HIV Vaccine Trial Network (HVTN) 505 (NCT00865566), a multi-center, placebo-controlled preventive HIV-1 vaccine efficacy trial conducted in the United States with 2504 participants enrolled was terminated for vaccine efficacy futility [3]. This study specifically assessed the safety and efficacy of a DNA prime and recombinant adenovirus serotype 5 (rAd5) boost vaccine regimen (DNA/rAd5) among cisgender men and transgender women who had sex with men. Despite lack of overall efficacy, sequence analysis of infecting strains conducted by deCamp et al. [4] showed greater distance between envelope glycoprotein (Env) sequences and the subtype B vaccine insert among vaccine- compared to placebo-recipients, suggesting that vaccination influenced susceptibility to at least some circulating strains.

Consistent with this hypothesis, multiple correlates of risk (CoR), measures or traits that relate to the acquisition status of trial participants, have been reported from this trial using a number of analytical approaches. With Cox proportional hazard models, Janes et al. [5] identified associations between Env-specific CD8+ immune responses and HIV-1 acquisition risk. Fong et al. [6] used immunogenicity data to model risk with a forward stepwise feature selection and modeling. Neidich et al. [7] performed a CoR analysis with both logistic regression and an ensemble classifier and reported several humoral correlates, including Env-specific IgG3, Env-specific

antibody binding to FcγRIIa, and antibody phagocytic activity. While further support of the biological validity of these factors could be found in associations between Env-specific antibody binding to FcγRIIa and viral loads among participants who did acquire HIV [7], limitations remain: early study termination resulted in a small number of cases, limiting statistical power, and immune markers tended to exhibit only low to moderate statistical significance between acquisition cases and controls that did not acquire HIV.

Similarly, the Antibody Mediated Prevention (AMP) passive vaccination trials, which tested the ability of the broadly neutralizing antibody VRC01 to prevent HIV acquisition, also failed to meet its primary endpoint criteria of reducing HIV acquisition [8]. However, differences in the sequences [9] and sensitivity [8] of infecting viruses to neutralization by VRC01 between active and placebo study arms have been interpreted as proof of concept for the ability of broadly neutralizing antibodies to provide protection from acquisition of sensitive strains. Indeed, efficacy for reduced likelihood of acquisition by the subset of circulating viruses that could be effectively neutralized by VRC01 was high (~75%). Further, the likelihood of acquisition was inversely proportional to the estimated level of antibody and neutralization titers present at the projected time of acquisition [10,11]. Lastly, evidence of efficacy can also be found in the observation of reduced viral load among breakthrough cases [12]. Thus, while some might consider analysis of CoR in a trial with such an "ineffective" intervention to be simply misguided, much was learned from such analysis in the AMP trials. Differences in the real and estimated power of different studies to capture overall efficacy and these more nuanced effects can be used to appropriately soften interpretation of such "failures". Indeed, these studies provide clear evidence as to the value of case-control analysis even when formal overall endpoint efficacy criteria have not been met.

CoR analysis is a mainstay of vaccine research, yet traditional case-control analytical approaches using acquisition status groups can present limited power to unveil correlates of protection (CoP). In particular, both high vaccine efficacy and low vaccine efficacy can pose analytical challenges: the number of cases may offer limited power, only a single binary outcome (e.g., infection) may be available, and there may challenges to identifying well-matched controls [13], as examples. Here we focus on experience from HIV-1 vaccine efficacy trials, in which acquisition rates are low, efficacy is low, and exposure status (and therefore actual protection status) cannot be determined. In such cases the class that did not experience acquisition is only a modest proxy for the protected class, and CoR analysis may fail to identify true CoP. While over-representation of controls is a long-standing and commonly used means to improve upon the ability to detect CoR [14,15], particularly in studies in which the number of cases is limited, additional analytical approaches may offer the potential to discover candidate correlates that otherwise may be missed.

In previous work, we have established proof of concept for the potential of semi-supervised machine learning (ML) approaches to contribute to discovery of novel candidate correlates of protection by using acquisition outcome and immunogenicity data to make inferences as to the protection status of study participants that did not acquire HIV [16–18]. Specifically, these ML methods aim to distinguish groups with and without HIV in the absence of definitive examples of the group that is protected from acquisition. In this application, positive and negative groups reflect unprotected and protected classes, respectively. Definitive class labels are only available for a subset of the unprotected group, those who were exposed to and acquired HIV. The lack of definitively labeled negative group examples, that is, demonstrably protected participants, points to the potential utility of Positive-Unlabeled (PU) inference methods. These methods have been previously explored to address this challenge using both simplified [16] and more complex synthetic immunogenicity data [17]. Coupled to permutation testing as a means to establish the analytical robustness of protection class inferences [18], these approaches have demonstrated improvements in the ability to discover true CoP in PU learning simulation scenarios generated from real-world immunogenicity profiles in which ground truth protection status has not been considered in making class inferences but has been available for validation. To this end, the success of PU learning-based modeling approaches to reveal true CoP that are missed both when acquisition status rather than inferred protection status groups are compared, even when multivariate modeling of acquisition status is performed [17], demonstrates the potential of employing this advanced ML method, despite its inherent bias, to improve our understanding of vaccine-induced

protection by reliably identifying the study participants with immunogenicity profiles most similar to those who have acquired HIV.

## Results

### Protection status inferences

Humoral and T cell immunogenicity data previously reported for 25 participants who acquired HIV after week 28 (cases) and 125 participants who did not acquire HIV over all follow-up (controls) from among vaccine recipients in the HVTN505 trial (Fig 1A) were analyzed [3,5,7]. Vaccine recipients who acquired HIV have been definitively exposed to the virus and are also demonstrably unprotected by the vaccine. In contrast, given the assumptions that the vaccine did have some, albeit small, protective effect and that exposure rates are low, the vaccine recipient group who did not acquire HIV is presumed to be a mixture comprising a minority of protected (if exposed, would not have acquired HIV) and a majority of unprotected (if exposed, would have acquired HIV) individuals. The PU learning pipeline employed seeks to infer the protection status of these unlabeled vaccine recipients without HIV (hereafter "unlabeled") based on defining the degree of similarity in their response profiles to the vaccine recipients with HIV, and then to use these newly inferred protection status class labels to identify candidate CoP (Fig 1B).

We sought to employ a state-of-the-art PU learning method, termed PU Bootstrapped Aggregation, or Bagging, to predict the control group participants' level of protection based on the degree of similarity of their immunogenicity profiles as compared to those observed among those who acquired HIV. By aggregating predictions from numerous classifiers built with positive class samples and bootstrapped unlabeled class samples, we computed a protection score for each unlabeled control group participant by averaging across their "out-of-bag" (OOB) predictions. A binary protection status classification boundary was then defined with or without knowledge as to the proportion of "protected" participants expected within the cohort using the rank of this score such as provided by the calculation of vaccine efficacy. Confidence in the robustness of these scores and their associated class inferences was evaluated using permutation tests [18]. Lastly, inferred protection status classes that met robustness testing criteria were then used to identify candidate CoPs. With the goal of CoP discovery in mind, and especially in cases where the same immunogenicity data is used to infer protection status as well as be evaluated as potential CoP, exceeding the performance baseline set by permutation testing is key to combating type I errors and over-optimism in identification of potential CoP expected to result from circular analysis.

Dimensionality reduction methods were used to visualize relationships between immunogenicity data and HIV acquisition status outcomes in an unsupervised fashion (Fig 2A). Visual inspection of serostatus labels following Principal Component Analysis (PCA) and t-distributed Stochastic Neighbor Embedding (T-SNE) showed some evidence of similarity among the positive case samples within the larger space observed for the control samples. This pattern suggested that semi-supervised approaches could have the potential to meaningfully differentiate among unlabeled samples that were more or less similar to the known positive class.

Using a previously described and validated PU Bagging approach [19], each sample was assigned a consensus score reflecting its likelihood of classification as an unprotected (positive) sample across repeated modeling (Fig 2B). Reassuringly, visual inspection of PU Bagging scores appeared concordant with expectations based on HIV acquisition status labels, however, a rigorous means to assess the robustness of these class inferences was needed. To this end, given the absence of ground truth protection status labels to validate PU Bagging score-based predictions, we employed a positive set evaluation method to serve as a confidence test for the robustness of prediction results. We used permutation testing to determine whether the degree of concordance obtained between scores of known positive samples yielded for the participants who acquired HIV exceeded scores that were observed when HIV acquisition status labels were permuted and permuted "known positive" class samples were drawn at random. While such a test cannot establish that class predictions are biologically correct, it can effectively evaluate whether they are statistically different than would be expected at random; that is, it allows us to establish confidence in whether the case group does or does not have a distinct immune response from members of the control group drawn by chance.

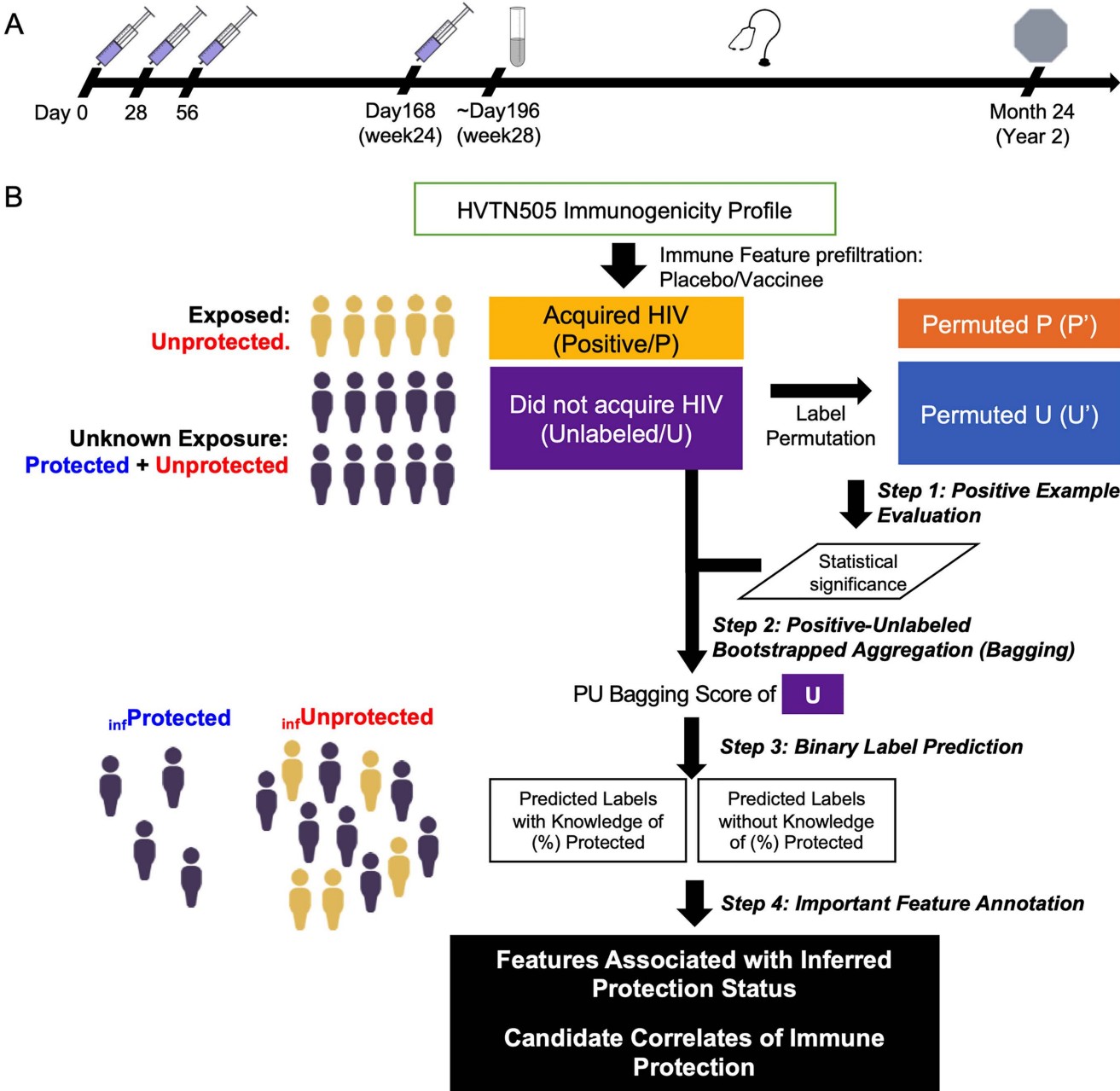

**Fig 1. Positive-Unlabeled Learning-based pipeline to infer protection status and identify candidate correlates of protection in HVTN505 participants. A.** Timeline of HVTN505 Trial. Study participants received placebo or a recombinant DNA plasmid vaccine injection on days 0, 28 and 56 and a recombinant adenoviral serotype vector vaccine injection on day 168. Immune responses to vaccination were assessed on week 28 and the clinical endpoint (HIV acquisition status) monitored between week 28 and month 24. **B**. Schematic of PU Learning approach to infer protection status. Vaccine recipients that have acquired HIV are definitively unprotected and represent known positives (P). Unlabeled (U), or vaccine recipients that have not acquired HIV are hypothesized to include both protected participants as well as unprotected participants who were simply not exposed. In Step 1, a PU Bagging classifier was used to define class probabilities for actual PU labels and permuted PU class labels. In Step 2, the PU Bagging scores for participants in the unlabeled (U) class were defined. In Step 3, these class scores were used to generate a binary protection status class inference. In Step 4, immunogenicity features that help classify inferred protection status class labels or show univariate associations with predicted class are identified as candidate correlates of protection.

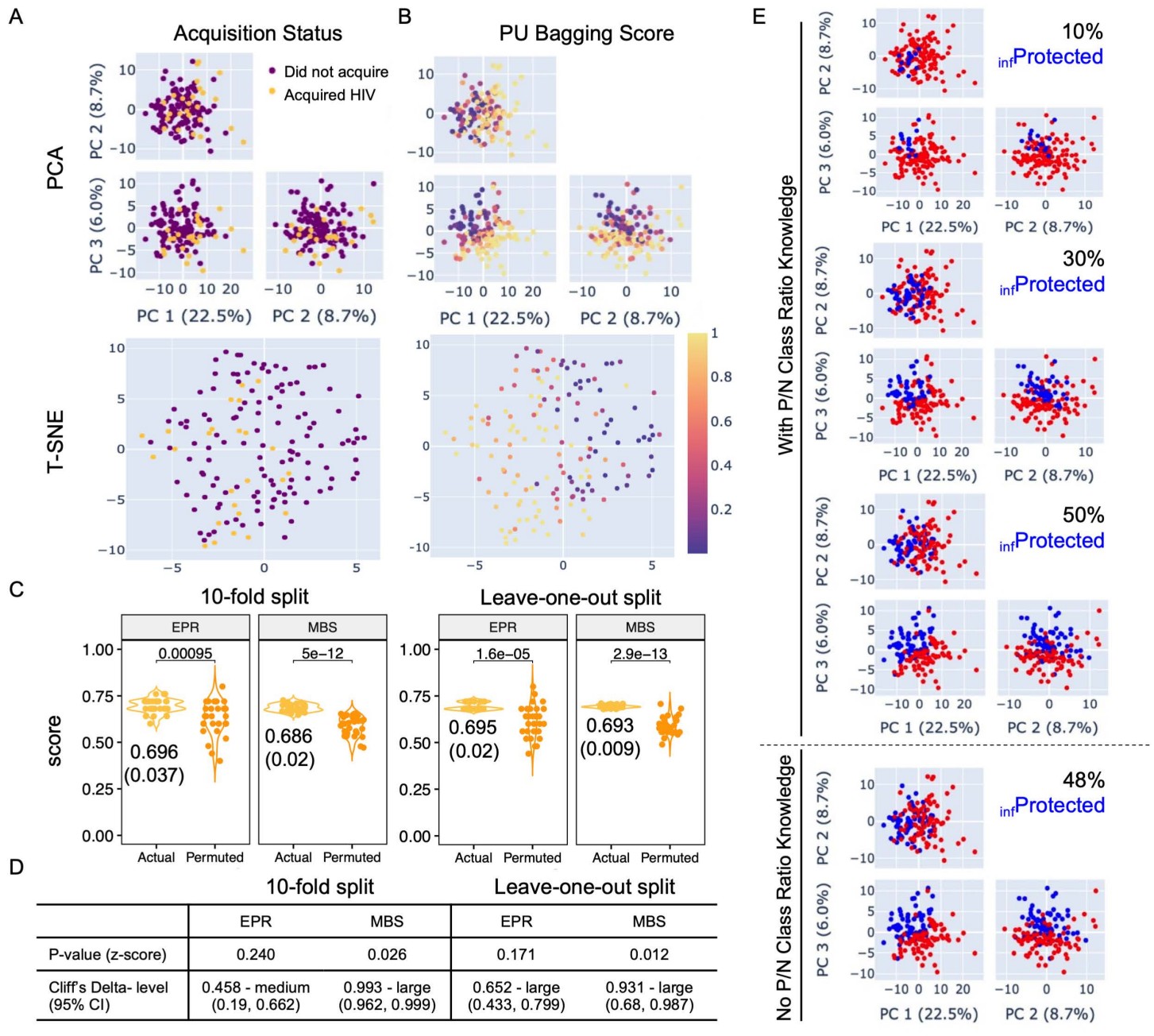

**Fig 2. Immunogenicity profiles relate to acquisition status and can be used to infer protection status. A-B.** Principal component analysis (PCA, top) and T-SNE (bottom) to visualize the global structure of the dataset. Samples are labeled according to the HIV acquisition status outcome **(A)** or inferred protection status score obtained from the PU Bagging algorithm **(B)**. **C.** Explicit Positive Recall (EPR, left) and Mean Bagging Score (MBS, right) value distributions were calculated for known positive class (Acquired HIV) samples for models trained on actual (yellow) and permuted (orange) class labels. Two sample independent T-test p values comparing score distributions for actual and permuted class labels are reported in inset. Mean and (standard deviation (SD)) of scores for the actual class labels are presented below each distribution. **D.** Statistical analysis comparing class probability scores of known actual and permuted positive samples. Statistical significance between score distributions for actual and permuted labels was defined by p-value from z-score, and effect size estimated using Cliff's Delta. **E.** PC biplots depicting classification of unlabeled samples with knowledge of the ratio between protected and unprotected subjects among the unlabeled class set to 10, 30, and 50% (from top), or without the benefit of knowledge of the proportion of positive and negative classes anticipated among unlabeled samples (bottom). Participants classified as infProtected are indicated in blue and infUnprotected in red.

We first compared the overall positive set PU Bagging scores under true acquisition outcome labels (actual) and those observed across multiple different permuted acquisition labels following label scrambling (permuted). For both repeated ten-fold and leave-one-out cross-validation runs, both explicit positive recall (EPR) and Mean Bagging Score (MBS) values for the actual positive samples typically exceeded those of permuted positive samples, providing evidence of a biological difference in immunogenicity profiles observed in vaccine-recipients who went on to acquire HIV during the trial's follow up period as compared to those who did not (Fig 2C). Consistent with previous work [17], for the true positive class, MBS values were more tightly distributed across replicates than were EPR scores under a 30-time repeated k-fold "spy positive" evaluation [20]. Variance in the distribution of these scores after label permutation was also greater. Differences between actual and permuted HIV acquisition status class labels were observed for both positive set prediction evaluation metrics across various choices of $K$ (Fig 2D). Two-sample independent T tests showed the expected inflation of significance associated with an increasing number of permutations; z-score p values were less susceptible to this artifact and sometimes but not always exceeded the conventional threshold of statistical significance. Effect size assessed using Cliff's Delta indicated a medium to large difference between actual and permuted positive class scores, providing further analytical basis for confidence in the potential biological relevance of protection status inferences based on reliable differentiation of the actual true positives (Fig 2D).

While PU Bagging scores provide a means to compare the relative differences in protection inference confidence, they don't provide an explicit boundary between classes. For cases in which the proportion of protected participants among all participants who did not acquire HIV are known or may be reasonably estimated, such as by overall vaccine accuracy, a percentile cutoff based on PU bagging score rank can be used to set the class threshold. Visualization of protection status class labels assigned based on different degrees of efficacy, or to a score set to 0.5, demonstrate how binary class assignments relate to the lead principal components in this study (Fig.2E). To characterize the reproducibility of PU scores, class scores for the unlabeled samples was characterized across 30 modeling iterations (S1 Fig in S1 File), demonstrating good robustness for scores that result from different folds.

## Relationships with behavioral risk

Additionally, an exploratory analysis was conducted to assess the correlation between clinical traits and acquisition outcome and the potential confounding effect between covariates among vaccinated participants in the studied cohort (S1 Table in S1 File). Consistent with what was observed in the entire cohort, an association was identified between behavioral vulnerability (OR = 4.32, p = 0.0216) and acquisition outcome among cases and controls, adjusted for other covariates (BMI, Race, Age) [3]. Intriguingly, a significant association between behavioral vulnerability and inferred protection status classified by the PU Bagging pipeline was not observed (OR = 1.76, p = 0.217) as either a univariate factor or in a multivariate logistic regression model. This result suggests that immunogenicity measurements possess the potential to forecast participants' expected protection level regardless of behavioral vulnerability factors that are correlated to the likelihood of HIV acquisition.

Next, we sought to leverage behavioral risk information to explore the validity of protection inferences, as behavioral vulnerability can be considered a proxy for HIV exposure. PU scores for participants with high behavioral risk were compared between those who did and did not acquire HIV (S2 Fig in S1 File). PU scores were significantly (p = 0.047) greater in high behavioral risk participants that acquired HIV as compared to those with high behavioral risk that did not acquire HIV. This result provides support for the potential biological validity of class predictions as it shows that inferred protection scores are associated with HIV acquisition among vaccine recipients with greater likelihood of exposure.

## Immunogenicity feature contributions to inferred protection status

To define the features that contribute to inferred protection status classes, we next trained a Random Forest (RF) classifier, a decision tree-based machine learning model, to predict membership in inferred protection ($_{inf}$Protection) and infection status classes based on immunogenicity data. Stratified five-fold cross-validation was used to fine-tune the

hyperparameters of the RF classifier and estimate the generalizability of the established model on unseen data [21]. As a pre-requisite for using these models to assess feature contributions, classification accuracy, as defined by area under the Receiver Operator Characteristic curve (ROC-AUC), was considerably greater for predictions of inferred protection status (Mean AUC: 0.94; SD: 0.03)) as compared to HIV acquisition status labels (Mean AUC: 0.66; SD: 0.04) (Fig 3A). This result suggests that the model has learned patterns and dependencies in the data that contributed to these class inferences, as is expected from its reliance on this data in training. Given that this circularity will drive over-optimism in classification performance, this high degree of accuracy should not be interpreted as validating the protection status predictions, but rather as simply a basis for confidence in the relevance of features used to model inferred protection status.

We next investigated the immunogenicity response features important to these classifications. As compared to prediction of acquisition status, which relied largely on both CD4 and CD8 T cell response attributes as well as IgG response magnitude, and to a lesser extent on virus-specific IgG3 levels, IgG3 features, supported by CD8 T cell features, and additionally IgA and antibody-dependent phagocytosis (ADCP) measurements contributed to classification of inferred protection status (Fig 3B). In order to evaluate the robustness of these contributions to predicting inferred protection status and to directly compare with contributions to acquisition status, we repeated classification modeling 100 times. As might be expected from their potential contributions to inferred protection status, importance scores for both features were significantly greater under the label of inferred protection as compared to that under acquisition outcome (Fig 3C), demonstrating a reproducible association between Env gp140 specific IgA and ADCP responses and vaccine-elicited immune protection as inferred by PU learning.

The robustness of these models was evaluated across 30 repeated cross-validation runs with area under the receive-operator curve (AUC) and balanced accuracies reported as outcome metrics. Inferred protection status class labels were more accurately predicted when actual rather than permuted labels were used in training models optimizing either overall or balanced accuracy (Fig 3D). In contrast, the accuracy of predictions of acquisition status using actual class labels was not always better than that achieved when models were trained on permuted acquisition status labels (Fig 3D). These results emphasize both the distinction between inferred protection status and acquisition status as well as the difficulty inherent to modeling acquisition status in studies where both vaccine efficacy and pathogen exposure rate is low and there is significant overlap between the immunogenicity profiles of many participants without HIV and those that were exposed and acquired HIV, resulting in natural confusion between them.

As an alternative to understanding contributions of combinations of immunogenicity response features to modeling acquisition and inferred protection status classes, we next evaluated associations between each feature and class assignments by Mann-Whitney U test, a non-parametric bivariate method that determines whether a statistically significant difference is observed between different groups of samples. In general, each of the roughly two dozen individual features that were significantly distinct between acquisition status groups exhibited the same directionality and greater confidence for inferred protection status groups, consistent with their playing a key role in driving protection status inferences (Fig 3E). Beyond Env-specific IgG3 and CD8+T cellular responses, which showed high importance and significance in both multivariate modeling and bivariate statistical tests for both acquisition and inferred protection classes, we observed increased univariate statistical significance in FcγRIIa- and FcγRIIIa-binding of Env-specific antibodies under the inferred protection as compared to HIV acquisition status class labels (Fig 3E-F). Consistent with the increased importance of ADCP and Env gp140–specific IgA responses observed in multivariate models, bivariate statistical significance was also seen in the Mann-Whitney U test (Fig 3D). Whereas ADCP activity was roughly two-fold higher among participants inferred to be protected, Env-specific IgA was roughly two-fold lower (Fig 3F). In total, roughly twice as many additional features exceeded the nominal significance threshold. Similar results were observed when adjusting for clinical traits (S3 Fig in S1 File).

A subset of features that were either previously reported as humoral CoR or newly identified here as inferred CoP based on their ability to predict or associate with inferred protection status are shown for the ConS gp140 specificity for individual participants based on their acquisition or inferred protection status (Fig 4A). As examples, differences in the

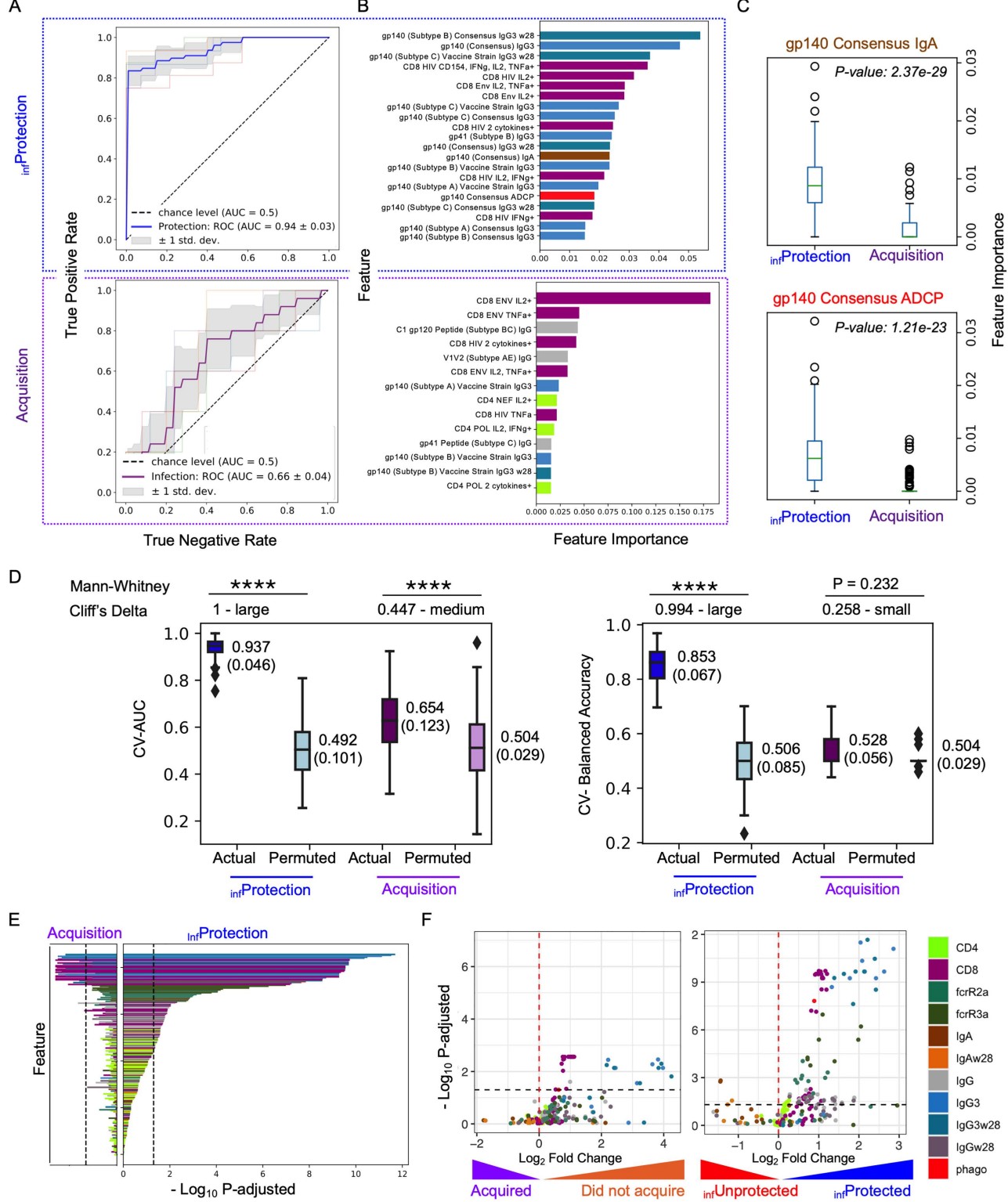

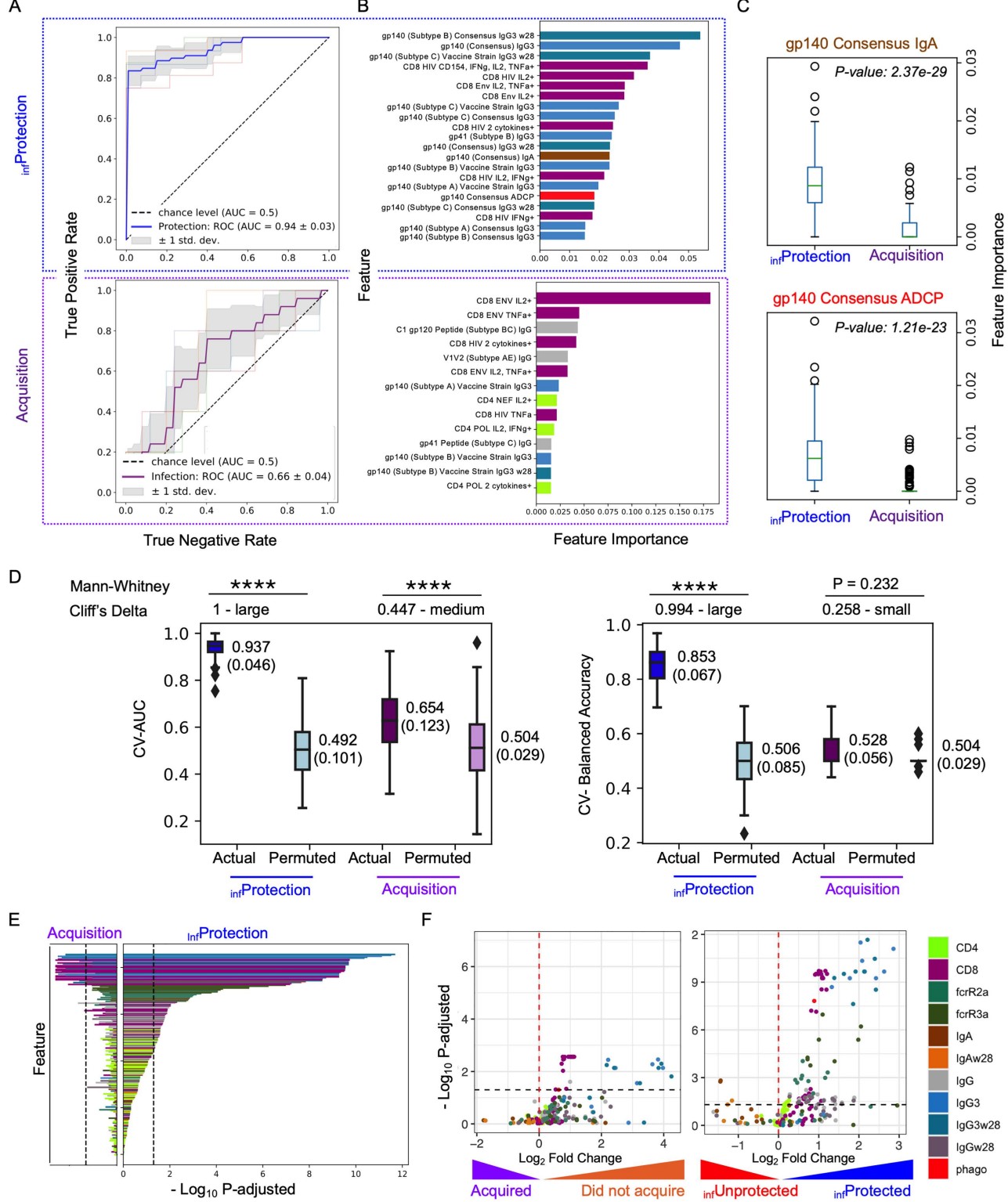

**Fig 3. Cross-validated multivariate modeling and permutation testing support the analytical robustness of inferred protection status. A.** Receiver Operator Characteristic (ROC) curves of a Random Forest classification model under stratified 5-fold cross validation to predict inferred protection (infProtection, top, in blue) or acquisition (bottom, in purple) status. Gray shaded area represents ±1 SD of the performance in each fold. Dotted line

indicates the performance expected at random, and area under the curve (AUC) values are reported in inset. **B.** Feature importance scores of the top ranked humoral response features, colored by feature type (legend at bottom right), for each classification task. **C.** Boxplots depicting feature importance scores for Con S Env gp140-specific IgA and ADCP for inferred protection and acquisition class labels in 100-time repeated modeling. Statistical significance was defined by Mann-Whitney U test. **D.** Boxplots of model performance assessed by permutation test. 30-time repeated cross-validated AUC (left) and Balanced Accuracy (right) under predicted protection level and acquisition label was compared to that under randomly shuffled labels for each (permuted labels). The multiple-time cross-validated performance was annotated with the format in Mean (Standard Deviation). Mann Whitney U test and Cliff's Delta Estimate (annotated on the top of the panels) were employed to determine the statistical significance between groups (****: p-value < 0.0001). **(E-F)** Identification of correlates. **E**. Mann-Whitney U tests were employed to compare each immunogenicity response feature for HIV acquisition status (left) and inferred protection status (right) class labels. Features are ordered on y-axis according to the rank of -Log10 p-adjusted obtained from protection label and colored by feature type. **F.** Volcano plots depicting confidence and magnitude of differences in immune response features (colored by type) associated with HIV acquisition (left) and inferred protection (right) status. The Benjamini-Hochberg procedure was used to adjust p-values from Mann Whitney U test to reduce the false discovery rate.

relationships between classes for ConS gp140-specific IgG and IgA for were pronounced (Fig 4B, for U statistics see S2 Table in S1 File): whereas a difference was not observed between acquisition status classes (p = 0.1 and 0.4 for IgG and IgA, respectively), there was high confidence in a group difference based on inferred protection status (p = 0.0029 and 0.00033). Consistent with the unrelated prior RV144 HIV-1 vaccine efficacy study [22,23], while IgG responses were elevated among participants inferred to have been protected, the opposite directionality was seen for IgA.

Expanding on this latter observation, given that IgA measurements in this study were bimodal, with a large number of values around zero, they were also treated as a categorical factor including two levels, IgA negative (log-transformed value = 0) and IgA positive (log-transformed value > 0). A logistic regression model was employed to determine whether an association was observed between Env gp140-specific IgA+ and IgA- responders and inferred protection class. Here, after adjusting for clinical traits (including age, BMI, behavioral vulnerability, and race), HIV-1 Env Con S gp140 IgA binding MFI were identified to be inversely correlated to inferred vaccine-mediated protection as both a continuous (OR = 1.17, p = 0.0012) as well as a categorical variable (OR = 2.78, p = 0.0035) (S3 Fig in S1 File). In contrast to inferred protection status, no significant association was observed with acquisition outcome, as previously reported [7]. Overall, this analysis demonstrates that PU learning-based inferences of protection status strengthened confidence in previously reported CoR and identified novel candidate CoPs suggested by their ability to contribute to or associate with predicted protection status.

## Discussion

HIV remains a significant public health concern, and efforts to develop a safe and effective vaccine against the virus continue despite setbacks. Vaccine concepts derived from strategies to induce protective immunity in association with T cell responses and neutralizing antibodies, as well as antibodies with protective effector functions modeled from either nonhuman primate protection studies or to recapitulate the result of the RV144 trial, have all failed to meet predefined efficacy criteria [24–27]. While promising efforts are underway to elicit MHC-E-restricted and networked T cell epitope responses, and to guide the maturation of neutralizing antibody responses [28–30], we might reasonably anticipate that these strategies will also prove only partially protective, and that methods to improve power to discover candidate CoP will hold value. To this end, while broadly neutralizing monoclonal antibodies provide robust protection against experimental viral challenges in nonhuman primates [31–33], their limits in the face of circulating viral diversity in humans were elucidated by the AMP trials, which also failed to demonstrate overall efficacy despite reducing the likelihood of acquisition of sensitive strains [8]. This history points to the significant challenges ahead and suggests the value of maximizing insights afforded by each human HIV-1 vaccine efficacy trial.

In previous work, we applied a PU learning-based analysis pipeline on multiple synthetic and real-world vaccine immunogenicity data sets in which protection status was known but partially blinded in advance of analysis as a means to stress test the approach in cases where predictions could be validated [17]. These analyses also directly compared the

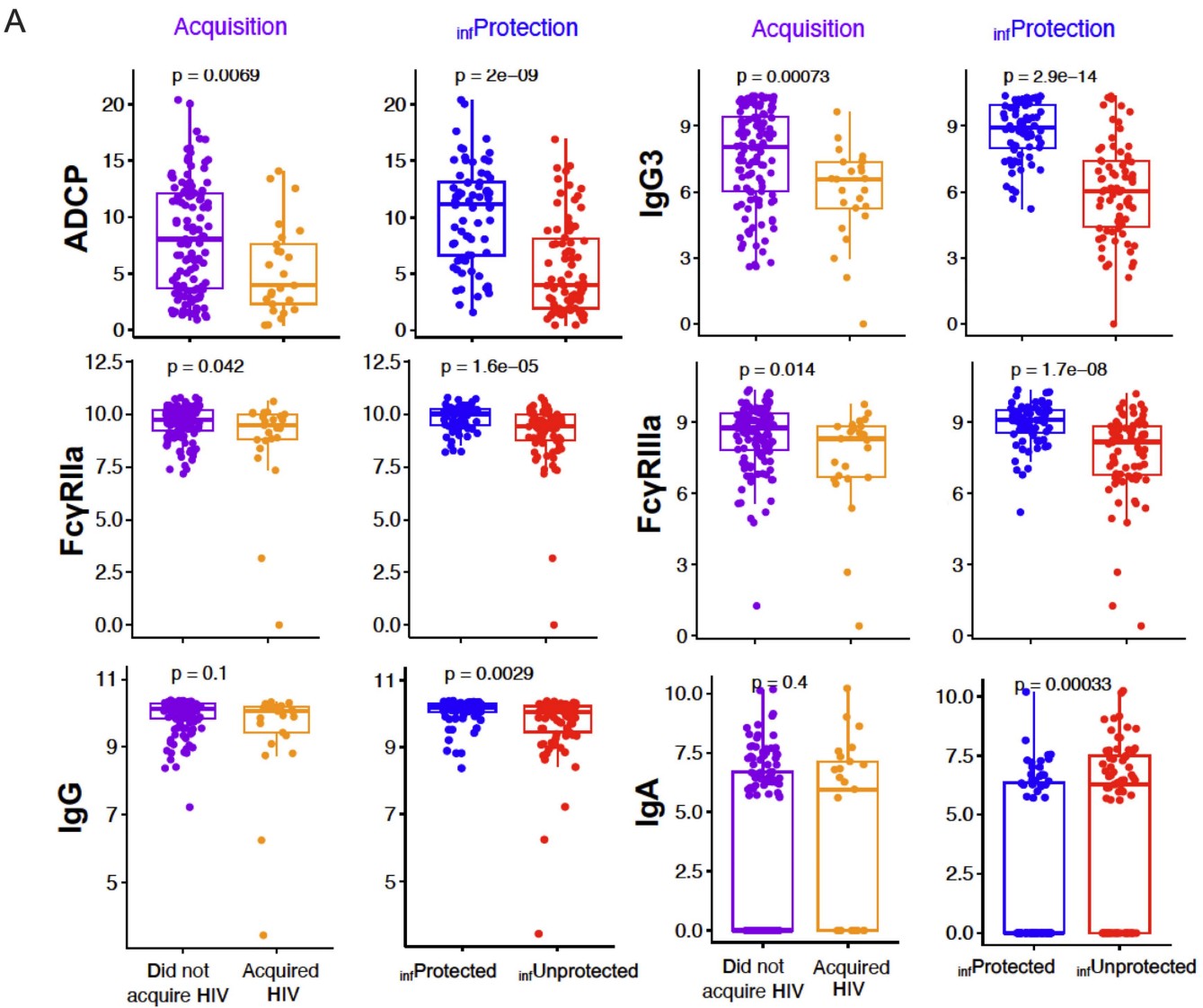

**Fig 4. Immunogenicity feature relationships with and contributions to models of acquisition and inferred protection status classes. A**. Box-plots of raw value distribution of notable ConS gp140 Env-specific humoral response measurements according to HIV acquisition and inferred protection (infProtection) status classes. Statistical significance defined by T test. **B**. Table of p values from Mann Whitney U test and Logistic regression model

| | Mann Whitney U test | | | | Logistic Regression Model | | | |
|---|---|---|---|---|---|---|---|---|
| | P-unadjusted | | P-adjusted | | P-unadjusted | | P-adjusted | |
| | Acquisition | infProtection | Acquisition | infProtection | Acquisition | infProtection | Acquisition | infProtection |
| **ADCP** | 0.00692 | 2.04e-09 | 0.0498 | 1.52e-08 | 0.00675 | 1.16e-07 | 0.0430 | 9.21e-07 |
| **IgG3** | 0.000728 | 2.90e-14 | 0.00732 | 3.24e-12 | 0.000905 | 7.99e-10 | 0.0151 | 9.94e-08 |
| **FcγRIIa** | 0.0421 | 1.64e-05 | 0.190 | **9.14e-05** | 0.00601 | 3.06e-05 | 0.0406 | 0.000171 |
| **FcγRIIIa** | 0.0141 | 1.66e-08 | 0.0950 | **1.09e-07** | 0.00498 | 1.26e-06 | 0.0370 | 8.00e-06 |
| **IgG** | 0.101 | **0.00294** | 0.322 | **0.0113** | 0.0160 | 0.00170 | 0.0774 | **0.00620** |
| **IgA** | 0.403 | **0.000327** | 0.628 | **0.00152** | 0.576 | **0.00120** | 0.742 | **0.00462** |

following adjustments based on clinical traits as covariates. P-unadjusted: p-values directly obtained from bivariate tests (left) or logistic regression models (right). P-adjusted: Benjamini-Hochberg method was used to reduce false discovery rate for multiple test hypothesis. Newly inferred candidate correlates are indicated in bold.

performance of fully supervised ML-based models to predict ground truth protection status after training to classify samples by acquisition status with semi-supervised models inferring protection status. In these comparisons, and bounded by the control for type I error provided by permutation testing, PU learning yielded better accuracy in correctly identifying protected individuals. In turn, as compared to those discovered through classification of acquisition status, these results also demonstrated identification of true CoP by PU learning that were not apparent from the partially blinded acquisition labels [17]. For HVTN 505, univariate analysis, classical logistic regression, and ML-based modeling identified a set of immunogenicity features related to acquisition outcomes [5–7]. Confidence in relationships between these features and outcomes was increased for inferred protection status classes; previously reported CoR, including Env-specific IgG3, ADCP, and Fcγ receptor binding of HIV-specific antibodies, which were associated with acquisition status [7], demonstrated more confident associations with inferred protection status. Additionally, this study highlighted that Env-specific IgA, which was not identified as correlated to HIV acquisition in HVTN 505 with case-control analysis based on acquisition outcome, contributed to multivariate models and exhibited statistical significance based on analysis of inferred protection status classes. Two of these features (Env gp140–specific IgA and ADCP), which were not observed among top features in models of acquisition status, are intriguing as there is support for their potential biological relevance to protection from infection based on relationships with acquisition outcomes for the RV144 trial [22,34–38], which tested a different regimen in a different population, as well as in preclinical studies [39–41] that preceded the Mosaico (HVTN 706/HPX3002) and Imbokodo (HVTN 705/HPX2008) trials. Antibody-mediated phagocytosis, which has the potential to contribute to clearance of both virions as well as infected cells, is potentiated by the IgG3 subclass [34,42] and leverages this activity of the FcγRIIa receptor [43]. Thus, the observation of relationships with inferred protection status for each of these features is consistent with their known biological connections and their potential biological relevance to reducing acquisition risk. In contrast to these features that were elevated among subjects that did not acquire HIV, or were inferred to be protected, envelope-specific IgA responses were correlated with increased relative risk of acquisition in the RV144 regimen [22], and higher inferred acquisition risk here. Envelope-specific IgA can interfere with the effector functions of envelope-specific IgG [23].

An alternative interpretation of these results is that while there are relationships among immunogenicity features related to their shared induction, none relate to vaccine-mediated protection from acquisition, because in failing to meet its endpoint efficacy criteria, the vaccine failed to definitively demonstrate that it provided protection. Indeed, none of these connections prove that these activities either correlate with or are mechanistic contributors to protection mediated by the HVTN505 regimen, nor do they establish that the protection status inferences made herein are correct. While they are at least consistent with expectations supported by antibody biology and the only HIV vaccine regimen to meet its overall efficacy criteria, there is a lack of means to externally validate these observations. Markers of exposure, specific data regarding actual behavioral risk rather than questionnaires, or acquisition outcomes following the formal observation period were not available to bolster or refute the potential accuracy of PU learning-based protection status inferences or the features associated with these predictions as candidate correlates. Further while the effect size was large for the comparison of positive class scores between actual and permuted input data, z-score p values were not significant for the EPR score and were marginal for MBS.

These results support the hypothesis that PU learning could "recover" markers of protection that are not identified by univariate, multivariate, or ML-based analysis of acquisition status—yielding the potential to contribute much needed insights about vaccine-mediated protection from HIV acquisition. While different experiments would be required to provide direct validation of either the inferred protection classes or the mechanistic relevance of novel candidate correlates, these

observations are supported by prior results as noted above. Nonetheless, validation of protection predictions in real-world data remain a limitation of this study due to the absolute absence of negative class examples representing protected participants in this clinical trial. Given this limitation, several methods were employed to test whether the classification outcomes may be concordant with the "ground truth" protection status. We first evaluated the positive-unlabeled learning method only with positive samples and compared the overall score obtained by this positive set with results under PU label permutation to control for type I errors in inferring protection status. Significant increases in positive class scores were observed under the actual HIV acquisition status labels compared to those observed following permutation, suggesting that immunogenicity profiles for participants who acquired HIV (true positives) were sufficiently distinct from at least some of the unlabeled samples as to have the potential to predict protection status among the participants without HIV. Furthermore, we studied the correlation between multiple previously described clinical traits to the classification outcome, including BMI, age, ethnicity, and behavioral vulnerability score. Promisingly, the significant correlation between behavioral vulnerability and acquisition outcomes was not observed for the inferred protection status classes, suggesting that while the likelihood of acquisition is related to behavior, protection status may not be. Supportively, among participants with high behavioral risk, which can be considered a proxy for HIV exposure, PU scores were associated acquisition.

On the other hand, inferred protection status is expected to be influenced by the set of feature inputs used. While feature sets that do not have reliable relationships with outcomes could be identified by permutation testing and used to reject overoptimistic protection inferences, it is likely that robust inferences supported by different data sets would nonetheless identify a non-identical set of samples or confidence values. To address this limitation, other data streams, such as transcriptomic profiles [44], could be employed for modeling. Hypothetically, concordance between protection status predicted by different sets and types of biological measurements would be observed among the same cohort of participants for accurate predictions, and use of one data stream for protection inferences and the other for candidate correlate discovery would be a "data thinning" approach that would more rigorously avoid the signal leakage inherent in the approach used here. Holding out features instead of samples would similarly permit separation of inferences of protection status from identification of candidate CoP, though immunogenicity data can exhibit a high degree of correlation. Each of these approaches is expected to decrease the rate of false positive observations, which despite permutation testing results that support the non-randomness of class inferences, can persist at the level of individual candidate CoP [17]. Because false positives can nonetheless be expected at some rate, caution needs to be paid when interpreting both the classification label inferred by the PU learning method as well as candidate CoP; whether an individual has truly been protected by the vaccine remains unknown.

PU learning makes several assumptions that are variably well-justified in this specific application. These assumptions include (i) that there are not false positive HIV test results, (ii) that the set of vaccine recipients that did not acquire HIV include both unprotected and protected participants, and (iii) that unprotected vaccine recipients have similar immunogenicity profiles whether they acquired HIV or did not. Among these, assumption (ii) may be questionable; because the HVTN505 regimen did not demonstrate overall efficacy, it is formally possible that there were no protected participants. However, the established identification of differences in immunogenicity data and infecting virus sequences between cases and controls [4–7] suggests otherwise. Assumption (iii) is sometimes referred to as the positive class being "selected completely at random". However, we know that the participants that acquired HIV were not a random subset of vaccine recipients but rather were enriched in individuals that presented higher behavioral vulnerability. In this case, the observation that inferred protection class scores were not related to behavioral vulnerability suggests that this assumption may be valid for models trained on immunogenicity data.

Overall, while the combination of machine learning approaches and advances in experimental approaches to generate large-scale, high-dimensional genetic and molecular profiles have already contributed to our existing understanding of the immune system, knowledge gaps remain. In this work, we intended to provide additional insight into the immune markers that may have contributed to the protection of some vaccinated participants in HVTN 505 by employing a reproducible PU

learning-based pipeline for candidate CoP identification with an integrated null-hypothesis comparison method. To the best of our knowledge, this is the first work to use real-world human trial case-control immunogenicity data to infer participants' protection levels with PU learning and face the limited ability to validate observations inherent to PU learning in this clinical setting. CoR previously identified by analysis of infection status using standard statistical methods exhibited increased statistical significance when analyzed based on inferred protection status, supporting the relevance of these CoR as potential CoP. Furthermore, this approach supported identification of additional features, concordant with multivariate ensemble models derived from acquisition status [7], as candidate CoP based on establishing their contributions to protection class inferences. These discoveries included CoR observed previously in other studies [22,34,36,37,45,46] , supporting the potential of PU learning to improve understanding of HIV vaccine-mediated protection and identify immune markers that may aid the design and development of an efficacious HIV vaccine. In sum, the protection models learned here raise hypotheses to test in future studies, and, with further experimental support, could be used to inform vaccine design and regimen optimization.

## Methods and materials

### Ethics statement

Participants in HVTN 505 (NCT00865566) provided written informed consent [3] for both the main study and this additional use of their samples. Analysis of HVTN 505 samples reported here was reviewed and approved through the Institutional Review Board at Dartmouth College.

### Datasets

This study utilized case-control immunogenicity and clinical information from 189 participants in the HVTN 505 clinical trial: 150 received a recombinant DNA plasmid vaccine injection followed by a recombinant adenoviral serotype vector vaccine injection, and 39 received placebo injections. HIV acquisition was assessed between Months 7 and 24. Vaccine recipients (N = 150) were labeled as "Positive" if HIV was diagnosed (N = 25) during the follow-up period, and unlabeled (N = 125) if they were not diagnosed (Fig 1A). These cases and controls were matched on a set of criteria as specified in Janes et al., [5].

### Immunogenicity features

As described by Neidich et al., Antibody-Dependent Cellular Phagocytosis (ADCP) [47], log-transformed multiplex Binding Antibody Multiplex Assay (BAMA) [48] and Fc Array assay [49,50] fluorescence intensities, and CD4 + & CD8 + T cellular responses [5] were assayed at 7 months after the first dose of vaccine [7]. To decrease false discoveries, a Wilcoxon test with Benjamini-Hochberg Procedure was performed on each immune feature to compare responses in placebo and vaccinated participants. Immunogenicity features for which $p > 0.05$ were excluded from further analysis based on lack of a vaccine-associated response.

### Clinical information

HVTN 505 study participants were restricted to cisgender men and transgender women who had sex with men. As described by Hammer et al. [3], the age, body mass index, ethnic information, and behavioral vulnerability scores were identified based on a weighted average of two variables from the ACASI questionnaires that were collected at baseline for each participant. Baseline characteristics of sample donors based on HIV acquisition status and inferred protection status groups are presented in S1 Table in S1 File.

### Positive unlabeled learning

Positive-unlabeled bootstrapped aggregation (Bagging) algorithms were first adapted to score the probability of the unlabeled samples being classified as "unprotected". We bootstrapped samples from the unlabeled class with matched

size to known positive samples and built a classifier to classify the remaining unlabeled samples. By iteratively bootstrapping different sets of unlabeled samples and predicting the rest of the unlabeled samples with established classifiers, the probability score of each unlabeled sample to be classified as "unprotected" (hidden positive) was calculated as the sum of its prediction when "out-of-bag" (OOB) divided by the number of iterations for which the sample was OOB. In this work, the number of total bootstrapped iterations was set to 200. A support vector machines (SVM) classifier with radial basis function (RBF) kernel, which demonstrated memory efficiency and effectiveness in high dimensional datasets, was used for the PU Bagging approach. To reduce the variance from a single-time prediction, the final PU bagging score for each unlabeled sample was averaged across 30 repetitions.

## Positive set evaluation

Because samples from definitively protected participants are not available to validate negative class predictions, confidence in predictions can only be assessed using the class of participants who acquired HIV. We therefore implemented testing to score the positive examples. Analogous to cross-validation, the samples that were labeled positive were randomly allocated into 5 folds. Each fold was rotated as "spy positive" samples and mixed with unlabeled samples. On each iteration, PU bagging SVM (RBF) was performed between the (k-1) folds of positive samples and unlabeled samples plus "spy positive" samples. Two scoring methods were utilized to assess the positive set using the PU bagging score of each positive sample as "spy positive". By treating samples with PU bagging scores ≥0.5 as positive, explicit positive recall (EPR) was adapted to score the positive set according to the proportion of replicates in which positive samples were correctly predicted.

$$Explicit\ Positive\ Recall(EPR) = \frac{N_{PredictedPositive}}{N_{Positive}}$$

$$Mean\ Bagging\ Score\ (MBS) = \frac{\sum scores\ from\ PU\ Bagging}{N_{Positive}}$$

An additional scoring method, termed Mean Bagging Score (MBS), was calculated by averaging the PU bagging score of all positive samples. In this work, 30 different fold splits of the positive set were performed under the actual label of infection.

## Permutation comparison

To establish a baseline comparison of the positive class scores (EPR and MBS) obtained, a permutation test was leveraged. In permutation comparisons, the class label of HIV acquisition was randomly shuffled multiple times, and the scores of the resulting permuted infected (P') set were calculated. In this work, the number of permutations was set to 30 or 100. Three statistical methods were employed to compare scores. Welch's T-test and Cliff's Delta statistics were applied to compare the scores for actual HIV serostatus labels as well as for permuted acquisition label. P-values of group z-scores were calculated between the sample mean of scores from actual labels and the distribution of scores from permuted labels. The z score was calculated as:

$$z = \frac{\mu - \mu 0}{\sigma}$$

Where $\mu$ denotes the mean score of positive sets based on 30 repetitions, while µ0 represents the mean score obtained by repeatedly permuting the labels and $\sigma$ refers to the standard deviation of the score distribution within the permutation

group, calculated using n-1 degrees of freedom. In this analysis, an upper-tailed test was conducted, assuming the hypothesis that $\mu > \mu 0$.

### Inferring label of varied protection level

Class scores were used to assign binary protection status class labels. Unlabeled samples were classified according to multiple cutoffs based on PU bagging score rank and the hypothetical level of protection (10%, 30%, 50% hypothesized protected). In the absence of information regarding the expected level of protection, binary protection status labels were inferred by aggregating the votes of the predictions from three previously described labeling strategies [17]: (1) Inferring binary class labels according to bagging score cutoff at 0.5, (2) Using 3-fold cross-validation to train a fine-tuned SVM classifier with known positive and matched number samples with lowest score rank as "reliable negative (RN)" samples, which was used to classify the rest of the unlabeled samples, (3) Training a biased classifier with known positive and RN ($N_{RN} = 25$), then, iteratively expanding the RN set by including the rest of the U that were predicted as "negative". In practice, the iteration was halted when no new negative samples were included or the number of iterations reached 20. Here, participants without HIV without PU Bagging Scores calculated were classified in the "unprotected" class by default.

### Supervised machine learning

Random forest classifiers were trained with all humoral response features with stratified 5-fold cross-validation for hyperparameter tuning. Centering and Scaling were used to process the immune features that presented different ranges of values. Features were ranked according to impurity-based feature importance under the label of inferred protection (i: protection) as candidate Correlates of Protection (CoPs) and under infection outcome as Correlates of Risk (CoRs). In this work, the top-ranked features with an importance score > 0.15 were shown. AUC and F1 scores under 5-fold cross-validation were employed to evaluate the generalizability of the model. In 100-time repeated modeling, the pooled importance score of selected features was compared between that under the acquisition outcome and inferred label of varied protection level. The model was implemented using the scikit-learn library (version 1.1.3) on Python 3.11 [51].

### Statistical analysis

Mann Whitney U tests were used to exclude the non-vaccine-elicited humoral responses from the feature set and test for the significance of immune features associated with infection or inferred protection status. Logistic regression models were built to assess the correlation between predicted outcome or clinical endpoint to immune features with covariates adjusted (age, BMI, race, behavioral risk score). The Benjamini-Hochberg procedure was performed to adjust for multiple hypothesis testing when immunogenicity features were compared between either acquisition or $_{inf}$Protection status groups. Consistent with convention, a p-value of 0.05, set prior to analysis, was used as a cutoff to determine statistical significance. Mann Whitney U tests were performed using SciPy library on Python 3.11 [52]; the Logistic regression model was implemented in R (version 4.2.2).

### Supporting information

**S1 File.** **S1 Table.** Clinical information for the vaccine recipients according to HIV acquisition status and inferred protection status outcomes. **S2 Table.** Mann-Whitney U statistics for selected immunogenicity features. **S1 Fig.** Score robustness. PU scores for the unlabeled class across modeling iterations. Symbol indicates mean and error bars indicate standard deviation. Samples are ordered from smallest to largest PU score. **S2 Fig.** Support for protection status inference validity. PU score for high behavioral risk participants who did not acquire HIV (purple), and those who did (orange). Welch's test p = 0.047. **S3 Fig.** Logistic regression to study relationships between immune features and prediction outcome after adjusting for clinical traits as covariates. A. Volcano plot of immune response features under HIV acquisition

(left) and inferred (i) protection status (right) outcomes from a logistic regression model. Regression model fit with (top) and without (bottom) adjustment for clinical covariates. The Benjamin-Hochberg procedure was used to adjust p-values from Mann Whitney U test to reduce false discovery rate. B. Plot of odds ratio values for each immunogenicity feature under HIV acquisition (left) and inferred protection (right) class labels. Error bars represent the 95% confidence interval (CI) and are colored according to feature type. Dotted line indicates no association.
(PDF)

## Acknowledgments

This study was supported in part by the Dartmouth International Vaccine Initiative.

## Author contributions

**Conceptualization:** Margaret E Ackerman.

**Funding acquisition:** Margaret E Ackerman.

**Investigation:** Shiwei Xu.

**Supervision:** Margaret E Ackerman.

**Visualization:** Shiwei Xu.

**Writing – original draft:** Shiwei Xu.

**Writing – review & editing:** Shiwei Xu, Aaron Hudson, Holly E Janes, Georgia D Tomaras, Margaret E Ackerman.

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
