## [Decision Letter · Decision Letter 0]

9 Mar 2025

PCOMPBIOL-D-24-01909

Expanded Insights into Correlates of Protection in the HVTN505 HIV-1 Vaccine Efficacy Trial Afforded by Positive-Unlabeled Learning

PLOS Computational Biology

Dear Dr. Ackerman,

Thank you for submitting your manuscript to PLOS Computational Biology. After careful consideration, we feel that it has merit but does not fully meet PLOS Computational Biology's publication criteria as it currently stands. Therefore, we invite you to submit a revised version of the manuscript that addresses the points raised during the review process.

Please submit your revised manuscript within 60 days May 09 2025 11:59PM. If you will need more time than this to complete your revisions, please reply to this message or contact the journal office at ploscompbiol@plos.org. Please include the following items when submitting your revised manuscript:

We look forward to receiving your revised manuscript.

Kind regards,

Jessica M. Conway

Academic Editor

PLOS Computational Biology

Denise Kühnert

Section Editor

PLOS Computational Biology

**Journal Requirements:**

At this stage, the following Authors/Authors require contributions: Shiwei Xu, Holly E Janes, Aaron Hudson, Georgia D Tomaras, and Margaret E Ackerman. Please ensure that the full contributions of each author are acknowledged in the "Add/Edit/Remove Authors" section of our submission form.

Potential Copyright Issues:

i) Figure 1. Please confirm whether you drew the images / clip-art within the figure panels by hand. If you did not draw the images, please provide (a) a link to the source of the images or icons and their license / terms of use; or (b) written permission from the copyright holder to publish the images or icons under our CC BY 4.0 license. Alternatively, you may replace the images with open source alternatives. See these open source resources you may use to replace images / clip-art:

6) Thank you for stating that "Code will be made available upon publication." Please note that, though access restrictions are acceptable now, your entire data will need to be made freely accessible if your manuscript is accepted for publication. This policy applies to all data except where public deposition would breach compliance with the protocol approved by your research ethics board. If you are unable to adhere to our open data policy, please kindly revise your statement to explain your reasoning and we will seek the editor's input on an exemption. Please be assured that, once you have provided your new statement, the assessment of your exemption will not hold up the peer review process.

7) Please amend your detailed Financial Disclosure statement. This is published with the article. It must therefore be completed in full sentences and contain the exact wording you wish to be published.

2) State what role the funders took in the study. If the funders had no role in your study, please state: "The funders had no role in study design, data collection and analysis, decision to publish, or preparation of the manuscript.".

**Reviewers' comments:**

Reviewer's Responses to Questions

**Comments to the Authors:**

**Please note that one of the reviews is uploaded as an attachment.**

Reviewer #1: In the present article, Xu et al. employ a Positive-Unlabeled (PU) inference method on immunogenicity data collected in the HVTN 505 study to infer vaccine-mediated protection scores against HIV acquisition by comparing patient immunogenicity profiles with those who acquired HIV. The authors then regressed these protection scores against the immunogenicity data to identify correlates of protection (CoP) against HIV infection. To test the robustness of their scores, the authors employed permutation testing, finding that their protection scores and CoPs were not artifacts of chance. These efforts confirmed previously reported correlates of risk (CoR) as well as identified novel CoPs, such as Env gp140-specific IgA and ADCP, that were not found through HIV acquisition-based models. Overall, this study is a well-reasoned application of novel machine learning methods to improve recognition of biological determinants in PU data that may also progress our understanding of CoPs in HIV vaccination. However, further efforts to verify the biological validity of the protection scores would greatly bolster confidence in PU inference methods and in the CoPs identified in this article. There are also several opportunities to improve the clarity of the figures and text.

Major Concerns

Figure 3A is missing the prediction lines.

It is claimed that there is some clustering in Figure 2A among positive samples, but is this really supported? Adding quantification of clustering separation, such as through the Silhouette score, would improve confidence in this claim.

The authors use permutation testing to validate their Positive-Unlabeled (PU) model captured immunological patterns different from those in randomized measurements. However, it remains unclear if the patients identified as protected from HIV infection were truly protected from infection or if they were never exposed to HIV. To their credit, the authors acknowledge that their permutation efforts cannot verify if the inferred protection statuses are biologically correct, and that such verification is not possible as patient HIV exposures are unknown. In the supplement, the authors conclude there is no significant association between behavioral vulnerability and inferred protection status, but that there is an association between behavioral vulnerability and HIV acquisition. As such, behavioral vulnerability could be used as a proxy for HIV exposure. Perhaps the biological validity of the PU-inferred protection may be bolstered by comparing inferred protection between acquisition and non-acquisition cases in patients with high behavioral vulnerability.

Minor Concerns

In line 68, the statement “While over-representation of controls is a commonly used means to improve upon the ability to detect CoR…” should include a citation to support its claim that oversampling is a common method in CoR.

In line 148, missing a “were” between “that” and “observed”.

Line 164: The authors allude to two-sample independent t-tests in Figure 2D, though the table in Figure 2D does not include t-test results. Similarly, it is unclear which statistical test was used to define significance differences in EPR and MBS means for Figure 2C.

The color for the lighter colored data points, such as yellow, should be adjusted so they can be seen more easily.

Figures 2A-B should be reorganized for improved clarity. Reorganizing into two columns, with each row representing a unique combination of principal components and each column corresponding to HIV acquisition or inferred protection, may better illustrate relationships between acquisition and inferred protection.

Figure 2B contains “nan” as a feature of importance. Is this a real feature? Can the authors include a dictionary of the features?

The triangles along the x-axis in Figure 3F should have their intersection at 0 log fold change, not the center of the plot.

In Figure 3B, the x-axis should be labeled, and a legend should be included to define what types of features each color represents; if these correspond to the legend in 3F, it would be beneficial to put the legend earlier with Figure 3B. Similarly, it is unclear which classification task each feature importance plot corresponds to. Presumably, these match the classification task labels in Figure 3A, but increasing the spacing between rows or adding some background shading across all figures corresponding to a classification task may improve clarity in this distinction.

Throughout Figure 4, the authors should report Mann-Whitney U test statistics alongside p-values when comparing acquisition and protection models; p-values should be used to define if a difference in means is significant, whereas the test statistics should be used to compare deviations in means across models.

Reviewer #2: This study analyzes data from HVTN 505, a Phase IIb HIV vaccine trial that failed to meet its efficacy criteria, using a novel machine learning approach called Positive-Unlabeled (PU) learning to gain new insights about vaccine-mediated protection. While the trial showed no overall efficacy, previous analyses suggested that some vaccine recipients may have been protected from certain viral strains. The researchers applied PU learning to infer protection status among vaccine recipients who didn't acquire HIV, allowing for improved detection of potential correlates of immunity. Using this approach, the study confirmed previously identified correlates of risk, such as vaccine-elicited anti-HIV-1 Env glycoprotein IgG3 antibodies and antibody-dependent phagocytosis, while also revealing new observations including a strong inverse correlation between vaccine-mediated protection and virus-specific IgA responses. The findings demonstrate the value of advanced analytical methods for extracting meaningful insights from failed vaccine trials, particularly in cases where traditional analysis methods may lack statistical power due to low efficacy and exposure rates. This analytical framework offers a new way to use case-control datasets to identify markers of effective immune responses even in the context of low overall vaccine efficacy.

Major Comments:

1. Circular Logic and Validation Concerns

- The paper uses the same immunological data both to infer protection status and then to identify correlates of protection, creating problematic circular logic.

- The validation approach using permutation testing is insufficient to prove the biological relevance of the inferred protection classifications.

2. Statistical and Methodological Issues

- The PU learning approach makes strong assumptions about the mixture of protected/unprotected individuals that are not well justified.

- Multiple testing corrections appear inadequate given the large number of features examined.

- The robustness of the protection status inferences across different modeling choices is not thoroughly evaluated.

- Inconsistent choice of k-fold from 3 (SVM), 5 (RF), to 10 (Bagging)!

- Inconsistent use of softwares, Mann Whitney U tests were performed using SciPy library on Python 3.11[47]; the Logistic regression model was implemented in R (version 4.2.2). Why not using Wilcoxon test in R instead of Mann Whitney U test in SciPy? Or if Python is a preferred programming language, why not using SciKit Learn to perform Logistic Regression?

3. Overstatement of Findings

- Claims about "discovering" new correlates of protection are overstated given the circular nature of the analysis.

- The biological plausibility and mechanistic relevance of the identified features is not adequately discussed.

- The limitations of inferring protection status without exposure data are downplayed.

4. Structural and Presentation Issues

- The methods section lacks sufficient detail about key modeling choices and parameters.

- Figures are not well presented and do not effectively communicate the key findings.

- The discussion does not adequately address alternative interpretations of the results.

- The practical utility for vaccine development is unclear given the methodological limitations

Additional Points:

- The manuscript requires significant editing for clarity and conciseness.

- Key controls and sensitivity analyses are missing.

- The statistical significance of many findings appears marginal.

- Important caveats and limitations are buried in the discussion.

The core idea of using machine learning to gain additional insights from vaccine trial data is interesting, but the current manuscript has fundamental flaws in its approach and overreaches in its conclusions. To be suitable for publication, the authors would need to:

1. Develop independent validation approaches that don't rely on circular logic.

2. More thoroughly evaluate the robustness of their findings.

3. Provide more rigorous statistical analyses.

4. Substantially improve the clarity of presentation.

Reviewer #3: See the attachment

**Have the authors made all data and (if applicable) computational code underlying the findings in their manuscript fully available?**

Reviewer #1: **No: ** I didn't see the raw data or a dictionary of the variable names. It's possible I just missed this.

Reviewer #2: **No: ** Only the datasets used in this analysis are available at

https://atlas.scharp.org/project/HVTN%20Public%20Data/HVTN%20505/begin.view.

Reviewer #3: None

PLOS authors have the option to publish the peer review history of their article (what does this mean? ). If published, this will include your full peer review and any attached files.

**Do you want your identity to be public for this peer review?** For information about this choice, including consent withdrawal, please see our Privacy Policy .

Reviewer #1: No

Reviewer #2: No

Reviewer #3: No

**Figure resubmission:**

**Reproducibility:**



---

## [Decision Letter · Decision Letter 1]

4 Nov 2025

Dear Dr Ackerman,

We are pleased to inform you that your manuscript 'Candidate Correlates of Protection in the HVTN505 HIV-1 Vaccine Efficacy Trial Identified by Positive-Unlabeled Learning' has been provisionally accepted for publication in PLOS Computational Biology.

Best regards,

Jessica M. Conway

Academic Editor

PLOS Computational Biology

Denise Kühnert

Section Editor

PLOS Computational Biology

To improve clarity, please point to the data dictionary clearly somewhere in your MS. The link provided takes the reader to a top-level list of folders associated with other publications, and it takes some hunting to find what's needed.

Reviewer's Responses to Questions

**Comments to the Authors:**

Reviewer #1: The authors have resolved my concerns.

Reviewer #2: The authors have answered my comments and the paper is now ready for publication.

**Have the authors made all data and (if applicable) computational code underlying the findings in their manuscript fully available?**

Reviewer #1: Yes

Reviewer #2: **No: **

PLOS authors have the option to publish the peer review history of their article (what does this mean? ). If published, this will include your full peer review and any attached files.

**Do you want your identity to be public for this peer review?** For information about this choice, including consent withdrawal, please see our Privacy Policy .

Reviewer #1: **Yes: ** Aaron S Meyer

Reviewer #2: No

---

## [Editor Report · Acceptance letter]

PCOMPBIOL-D-24-01909R1

Candidate Correlates of Protection in the HVTN505 HIV-1 Vaccine Efficacy Trial Identified by Positive-Unlabeled Learning

Dear Dr Ackerman,

I am pleased to inform you that your manuscript has been formally accepted for publication in PLOS Computational Biology. Your manuscript is now with our production department and you will be notified of the publication date in due course.

With kind regards,

Judit Kozma
